# Structural differences between C-terminal regions of tropomyosin isoforms

Małgorzata Śliwińska and Joanna Moraczewska

Institute of Experimental Biology, Kazimierz Wielki University, Bydgoszcz, Poland

## ABSTRACT

Tropomyosins are actin-binding regulatory proteins which overlap end-to-end along the filament. High resolution structures of the overlap regions were determined for muscle and non-muscle tropomyosins in the absence of actin. Conformations of the junction regions bound to actin are unknown. In this work, orientation of the overlap on actin alone and on actin–myosin complex was evaluated by measuring FRET distances between a donor (AEDANS) attached to tropomyosin and an acceptor (DABMI) bound to actin's Cys374. Donor was attached to the Cys residue introduced by site-directed mutagenesis near the C-terminal half of the overlap. The recombinant alpha-tropomyosin isoforms used in this study – skeletal muscle skTM, non-muscle TM2 and TM5a, and chimeric TM1b9a had various amino acid sequences of the N- and C-termini involved in the end-to-end overlap. The donor-acceptor distances calculated for each isoform varied between 36.4 Å and 48.1 Å. Rigor binding of myosin S1 increased the apparent FRET distances of skTM and TM2, but decreased the distances separating TM5a and TM1b9a from actin. The results show that isoform-specific sequences of the end-to-end overlaps determine orientations and dynamics of tropomyosin isoforms on actin. This can be important for specificity of tropomyosin in the regulation of actin filament diverse functions.

## INTRODUCTION

Tropomyosins are a family of two-chain coiled coil proteins and are regarded as actin "gatekeepers", which control access of numerous actin-binding proteins to actin filaments (*Gunning, O'Neill & Hardeman, 2008*). Tropomyosin (TM) binds cooperatively to actin and forms long chains along both sides of the filament due to end-to-end overlap between adjacent molecules. Binding of TM to actin involves weak but specific electrostatic interactions between periodic actin-binding sites on TM's coiled-coil and residues exposed on actin subunits (*Barua, Pamula & Hitchcock-DeGregori, 2011*; *Li et al., 2011*). In this closed state, TM inhibits activation of actomyosin ATPase at low myosin concentrations (*Lehrer & Morris, 1982*). Strongly bound myosin heads (S1) cooperatively shift the filament into the open state, which is associated with an azimuthal shift of TM away from the position occupied in the closed state (*Lehman & Craig, 2008*). The S1-induced shift from

Corresponding author
Joanna Moraczewska,
moraczjo@ukw.edu.pl

**Peer**J

the closed to the open state is a universal mechanism of actin filament activation executed in the presence of muscle and non-muscle TM isoforms.

TM isoforms are generated by several genes (four in vertebrates), selection of alternative promoters, and alternative splicing of the transcripts. In $\alpha$-tropomyosin, the products of the *TPM1* gene, the N- and C-terminal regions are encoded respectively by two (1a and 1b) and four (9a–d) alternative exons. Selection of the alternative promoter gives rise to high molecular weight (HMW) and low molecular weight (LMW) isoforms of TM. HMW tropomyosins bind along seven actin subunits, whereas LMW isoforms along six actin subunits. The main structural difference between these two TM types is the N-terminal sequence, which is encoded by exon 1a or 1b in HMW and LMW isoforms, respectively (*Lees-Miller & Helfman, 1991*).

Structures of tropomyosin intermolecular junctions were studied with the use of model peptides imitating sequences of the end-to-end overlaps. Peptides with sequences of skeletal TM encoded by exons 1a and 9a (*Greenfield et al., 2006*; *Murakami et al., 2008*), non-muscle TM encoded by exons 1b and 9d (*Greenfield, Kotlyanskaya & Hitchcock-DeGregori, 2009*), and smooth muscle TM encoded by exons 1a and 9d (*Frye, Klenchin & Rayment, 2010*) were analyzed to obtain NMR and X-ray structures. Although the structures differ in the number of amino acids forming the overlap and in specific interactions between amino acid chains, the three complexes are similar – the two $\alpha$-helical chains of the C-terminus spread apart and interlock with the N-terminal coiled coils. The structures revealed, however, a slightly different tilt of the axis of the C- and N-terminal coiled coil (*Frye, Klenchin & Rayment, 2010*).

To understand the molecular mechanisms controlling numerous functions of actin, high-resolution structures of F-actin in complex with different TM isoforms are required. Models of actin complexes with skeletal and cardiac muscle TMs in different activation states are available (*Barua et al., 2013*; *Barua, Pamula & Hitchcock-DeGregori, 2011*; *Barua et al., 2012*; *Behrmann et al., 2012*; *Li et al., 2011*; *Miki et al., 2012*). The models show that conserved residues repeating along TM form periodic interactions with charged or hydrophobic residues exposed on actin (*Barua, Pamula & Hitchcock-DeGregori, 2011*; *Li et al., 2011*). When bound to actin alone, the N-terminal residues Lys6 and Gln9, located with the overlap region, make important interactions with Asp25 on actin. On the other hand, the C-terminal half of the overlap does not contribute directly to actin binding. This region may be important for regulation of myosin interactions with the filament, but there is no experimental data supporting this hypothesis.

In our earlier work, we used steady-state Förster resonance energy transfer (FRET) to determine apparent distances between donors specifically attached to the N-terminal regions of different TM isoforms. Our data suggested that, in closed and myosin-induced open states the N-terminal segments of tropomyosin isoforms are differently oriented on F-actin (*Śliwińska et al., 2011*). However, the results did not provide any insight into the position of C-terminal segments of the studied isoforms. We do not know whether the regions located in the close vicinity to the end-to-end junctions are stiff or rather

flexible. Various flexibilities within this region of TM might be an important determinant of differences between TM isoforms in regulation of actin–myosin interactions.

The aim of this study was to analyze the positions of the C-terminal regions, adjacent to the end-to-end overlap, in four tropomyosin isoforms. Steady-state FRET between a donor attached to the Cys residue introduced into the C-terminal segment of each isoform and acceptor bound to actin's penultimate Cys374 was used. The data shows that localization of the C-terminal region in relation to actin's C-terminus is unique for each type of the studied isoform. Specific conformational changes associated with activation of the filament by strongly bound myosin heads suggest that depending on the sequence, the end-to-end overlap regions have different flexibilities.

## MATERIALS AND METHODS

Chicken skeletal muscle $\alpha$-actin, chicken skeletal myosin subfragment 1 (S1) and recombinant rat $\alpha$-tropomyosin isoforms were used. TM2, TM5a and TM1b9a were obtained as described in *Śliwińska et al. (2011)*. Recombinant skTM was modified by insertion of AlaSer at the N-terminus to compensate for low actin affinity of recombinant skTM due to the lack of N-terminal acetylation as described in *Robaszkiewicz et al. (2012)*. The Department of Biochemistry and Cell Biology is authorized by the Minister of the Environment (Poland) for laboratory use of genetically modified organisms (permit no. GMO: 01-112/112).

PCR-based oligonucleotide-directed mutagenesis (Stratagene) was used to create an attachment site for a fluorescent probe at the C-terminal region of TM. First, cDNA encoding all TM isoforms used in this study was mutated to replace the single Cys codon for Ser. The procedure was described in *Śliwińska et al. (2011)*. Then cDNA from the first stage of mutagenesis was used to create Ala269Cys-skTM, Ser269Cys-TM2, Ala232Cys-TM1b9a and Ser232Cys-TM5a mutants. The oligonucleotides used at this stage were the following:

9a Ala269(232)Cys: 5′-ctgaagtacaagtgcatcagcgaggagctggaccacg-3′
9d Ser269(232)Cys: 5′-gccaaagaagaaaacctttgcatgcaccagatgctggac-3′

All primers were synthesized and HPLC purified by the Laboratory of DNA Sequencing and Oligonucleotide Synthesis, Institute of Biochemistry and Biophysics, Polish Academy of Science (Warsaw, Poland).

Tropomyosin mutants were labeled with AEDANS and actin was labeled with DABMI according to the procedure described previously (*Śliwińska et al., 2011*).

Tropomyosin affinity for actin was measured in a co-sedimentation assay. Increasing amounts of tropomyosin (final concentrations varied between 0 and 10 μM) were mixed with actin (5 μM) in F-buffer: 2 mM HEPES, pH 7.6, 40 mM NaCl, 5 mM $MgCl_2$ at 22°C and ultracentrifuged. Protein content in supernatants and pellets was analyzed electrophoretically on SDS-PAGE as described in *Skórzewski et al. (2009)*.

The activity of actomyosin ATPase was measured in F-buffer. Myosin S1 concentration was 0.8 μM, F-actin was 9.2 μM, and tropomyosin was 1.2 μM. The reaction was started by addition of MgATP to 5 mM and stopped after 10 min with 3.3% SDS and 30 mM

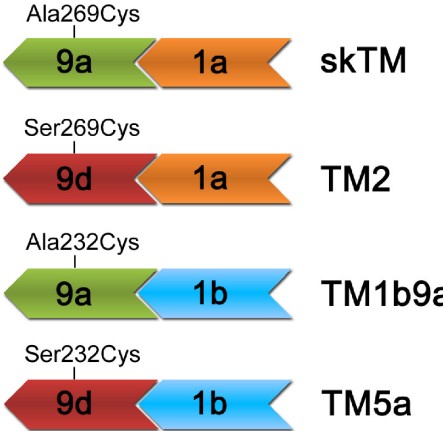

**Figure 1 Schematic illustration of the types of end-to-end junctions present in TM isoforms used in this work.** C-terminal sequences encoded by exon 9a (green) or 9d (red) overlapped with N-terminal sequences encoded by exons 1a (orange) or 1b (blue). The residues in C-terminal segments, which were mutated to cysteines, are shown.

EDTA. The amount of liberated phosphate was measured colorimetrically according to the method described in *White (1982)*.

Fluorescence anisotropy, FRET experiments, calculation of Förster critical distance ($R_0$), and donor-acceptor distance ($R$) were conducted according to the methods described in *Śliwińska et al. (2011)*.

# RESULTS

## Rationale for the labeling-site selection and characterization of the labeled tropomyosin isoforms

End-to-end overlap sequences present in the studied TM isoforms were formed by two N-terminal and two C-terminal variants of alternative sequences of rat $\alpha$TM. A schematic illustration of the overlap regions is shown in Fig. 1. In skeletal muscle $\alpha$TM (skTM), non-muscle isoforms (TM2, TM5a), and chimeric TM1b9a, an isoform that has no natural counterpart, N-termini encoded by exons 1a or 1b formed complexes with C-termini encoded by exons 9a or 9d. In order to create a specific reactive site for the attachment of a fluorescent probe to the C-terminal regions of tropomyosin isoforms, residue 269 in HMW isoforms or its counterpart in LMW isoforms (residue 232) was changed to Cys (Table 1). This residue was selected for the following reasons: (a) it is located at the outskirts of the end-to-end junction (*Frye, Klenchin & Rayment, 2010*; *Greenfield et al., 2006*; *Greenfield, Kotlyanskaya & Hitchcock-DeGregori, 2009*); (b) it is not involved in any interactions within the studied end-to-end complexes (*Frye, Klenchin & Rayment, 2010*); and (c) it is located on the outer surface of the two-chain coiled-coil tropomyosin (position *c* of the coiled-coil heptapeptide repeat), which allows for free motions of the attached label. Thus we expected that the attached label was sensitive to conformational differences within the end-to-end overlap.

**Table 1 Localization of cysteine mutation sites in the sequences of the C-termini encoded by exons 9a and 9d.** The upper row shows positions of the amino acid residues in the coiled-coil heptapeptide repeat. Ala or Ser residues changed into Cys to create AEDANS attachment sites are in red.

| | *f* | *g* | *a* | *b* | *c* | *d* | *e* | *f* | *g* | *a* | *b* | *c* | *d* | *e* | *f* | *g* | *a* | *b* | *c* | *d* | *e* | *f* | *g* | *a* | *b* | *c* | *d* |
|---|---|---|---|---|---|---|---|---|---|---|---|---|---|---|---|---|---|---|---|---|---|---|---|---|---|---|---|
| **9a** | D | E | L | Y | A | Q | K | L | K | Y | K | A/C | I | S | E | E | L | D | H | A | L | N | D | M | T | S | I |
| **9d** | E | K | V | A | H | A | K | E | E | N | L | S/C | M | H | Q | M | L | D | Q | T | L | L | E | L | N | N | M |

**Table 2 Functional properties of wild type and mutant tropomyosin isoforms labeled with AEDANS.** Conditions: 2 mM HEPES, pH 7.6, 40 mM NaCl, 5 mM MgCl$_2$ at 22°C. The numbers are average values ±S.E. taken from 2 to 9 independent experiments.

| TM isoform | $K_{app}$ ($\times 10^7$ M$^{-1}$) | Actin–myosin S1 ATPase activity (nmol Pi/mg S1/min) |
|---|---|---|
| skTM | 1.63 ± 0.57 | 103.0 ± 8.0 |
| skTM/A269C | 1.80 ± 0.26 | 103.8 ± 7.0 |
| TM2 | 2.25 ± 0.76 | 91.7 ± 13.0 |
| TM2/S269C | 2.04 ± 0.45 | 112.0 ± 10.0 |
| TM5a | 1.42 ± 0.36 | 169.0 ± 16.0 |
| TM5a/S232C | 1.49 ± 0.58 | 183.8 ± 20.0 |
| TM1b9a | 1.27 ± 0.40 | 108.2 ± 9.0 |
| TM1b9a/A232C | 1.24 ± 0.44 | 125.5 ± 9.0 |

Replacement of Cys for Ser in the central region of TM did not affect the basic functions such as actin binding and regulation of actomyosin ATPase activity (*Śliwińska et al., 2011*). In order to check whether these functions were conserved in TMs with Cys mutations in the C-terminal region, we measured actin binding regulatory functions of all studied AEDANS-labeled TM isoforms. Binding constants ($K_{app}$) obtained in the co-sedimentation assay show that all TM mutants bound to actin with high affinity (Table 2), consistent with previous work (*Moraczewska, Nicholson-Flynn & Hitchcock-DeGregori, 1999*). The mutations also did not significantly change the interactions between actin and myosin heads. In the presence of wild-type and mutant TMs the activities of actin–myosin S1 ATPase were similar (Table 2).

To check whether all isoforms of AEDANS-labeled TM bound to DABMI-labeled actin stoichiometrically, TMs were titrated with increasing concentrations of actin. The titration curves shown in Fig. 2 indicate that the fluorescence was maximally quenched at the ratio of TM to F-actin, that was close to stoichiometric. For skTM and TM2 the maximal quenching was reached at 1 TM to 6–7 actins and in the case of TM5a and TM1b9a at 1 TM to 4–5 actins. The results confirm high actin affinity of TM isoforms labeled in the C-terminal segment. Additionally, the titration curves show that in case of the isoforms with the 9d-encoded C-terminus (TM2 and TM5a), the maximal quenching of AEDANS was reached at a lower TM/actin molar ratio than in the case of their 9a-encoded counterparts (skTM and TM1b9a). As fluorescence quenching was due to FRET, the

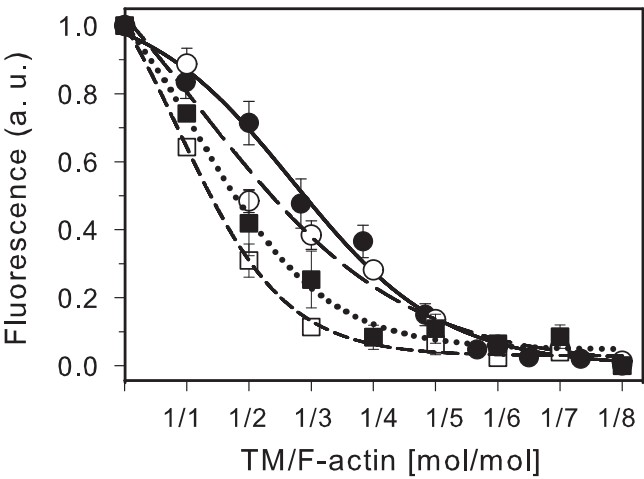

**Figure 2 Quenching of AEDANS-TM fluorescence by DABMI-actin.** sk TM (closed circles, solid line), TM2 (open circles, long dash), TM5a (open squares, short dash) and TM1b9a (closed squares, dotted line) at 0.6 μM were titrated with DABMI-actin. The data was normalized using the equation: $(I - I_{min})/(I_{max} - I_{min})$, where $I_{max}$ is the minimal intensity of the fluorescence of AEDANS-TM alone and $I_{min}$ is the minimal fluorescence intensity obtained in the presence of DABMI-actin. The lines were generated by fitting the experimental points to the ligand-binding equation in Sigma Plot. Conditions: 2 mM HEPES, pH 7.6, 40 mM NaCl, 5 mM MgCl$_2$ at 22°C. Excitation and emission wavelengths were 340 nm and 495 nm, respectively. The points were averaged from 3 to 5 independent experiments.

observed differences suggest that the C-terminal sequence encoded by exon 9d facilitates interactions of the C-terminal regions of TM2 and TM5a with actin.

## Distances separating the donor attached to the C-terminal region of tropomyosin from the acceptor bound to actin

Random orientation of the labels is important for FRET distance measurements (*Lakowicz, 1999*). To ensure that the probe was flexibly attached to Cys269/232 in all studied TM isoforms, we measured AEDANS fluorescence anisotropies. As shown in Table 3, the obtained anisotropies were low, and therefore we concluded that the label bound to free TM as well as to TM-actin complex in the presence and absence of S1 was randomly oriented. Thus, the orientation of the donor did not limit the FRET distance measurements.

Quantum yield of the donor ($Q_D$) and spectral overlap between emission of the donor and absorption of the acceptor ($J$) affect donor-acceptor critical distance ($R_0$) (*Lakowicz, 1999*). Because the local environment surrounding AEDANS bound to Cys269/232 could influence both parameters, $Q_D$ and $J$ were determined for each labeled TM mutant in the presence of unlabeled actin. The results show that the label attached to the C-terminal segment of the studied TM isoforms was exposed to different environments, which caused variations in $Q_D$, small shifts in $J$ and, in consequence, differences in $R_0$ (Table 4).

The FRET efficiency ($E$) was calculated from the fluorescence intensity of AEDANS-labeled TM in the absence and in the presence of the acceptor (TM saturated with DABMI-F-actin). The fluorescence of the donor was corrected for the increase caused

**Table 3 Anisotropy of AEDANS bound to C-terminal cysteine residues of TM isoforms.** Conditions: 0.6 μM AEDANS-labeled TM alone and with 4.8 μM actin ±5 μM S1 in 2 mM HEPES, pH 7.6, 40 mM NaCl, 5 mM MgCl$_2$ at 22°C. Excitation and emission wavelength were 340 nm and 495 nm, respectively. Average values ±S.E. were taken from 3 to 8 independent measurements.

| TM isoform | TM alone | TM-actin | TM-actin-S1 |
| --- | --- | --- | --- |
| skTM/A269C | 0.073 ± 0.008 | 0.086 ± 0.008 | 0.123 ± 0.012 |
| TM2/S269C | 0.072 ± 0.002 | 0.106 ± 0.005 | 0.137 ± 0.004 |
| TM5a/S232C | 0.066 ± 0.003 | 0.096 ± 0.004 | 0.124 ± 0.004 |
| TM1b9a/A232C | 0.095 ± 0.008 | 0.104 ± 0.003 | 0.127 ± 0.009 |

**Table 4 Spectral parameters of FRET between AEDANS-labeled tropomyosins and DABMI-labeled actin.** Conditions are as given in Table 3. Average values ±S.E. were taken from 10 to 14 independent experiments.

| TM isoform | $Q_D$ | $J$ ($\times 10^{14}$ nm$^4$ M$^{-1}$ cm$^{-1}$) | $R_0$ (Å) | $E$ | $R$ (Å) |
| --- | --- | --- | --- | --- | --- |
| skTM/A269C | 0.14 | 6.838 | 34.7 | 0.40 ± 0.01 | 37.3 ± 0.21 |
| TM2/S269C | 0.08 | 6.924 | 32.1 | 0.32 ± 0.01 | 36.4 ± 0.21 |
| TM5a/S232C | 0.20 | 7.012 | 37.1 | 0.32 ± 0.03 | 42.7 ± 1.20 |
| TM1b9a/A232C | 0.12 | 7.087 | 34.3 | 0.13 ± 0.02 | 48.1 ± 1.45 |

by the binding of unlabeled actin (3%–7% depending on TM isoform). The efficiencies obtained for the studied TM isoforms were used for calculations of the apparent distances ($R$) separating donor and acceptor. All calculated FRET parameters are collected in Table 4. The results suggest isoform-specific localization of the overlap region. The differences between the distances obtained for HMW isoforms (skTM, TM2) were small, which indicates similar position of the donor in relation to the acceptor. In LMW isoforms (TM5a and TM1b9a) the donor-acceptor distances were larger. It is worth noting that the distance obtained for TM1b9a was far from $R_0$, thus the sensitivity of FRET measurements was limited and the result can only be regarded as a rough estimate.

## Changes in tropomyosins' C-termini as an effect of myosin S1 binding to actin

Strong binding of myosin heads (S1) to the filament increases affinity of TM to actin and induces an azimuthal shift of TM chains. This changes the TM interactions with actin and activates the filament allowing for actin–myosin cross-bridge cycling (*Lehman & Craig, 2008*; *Moraczewska, 2002*). Saturation of the filament with myosin S1 caused a significant increase of the fluorescence of AEDANS bound to C-terminal segments of all studied TM isoforms. The average increase was about 14% and 18% for skTM and the non-muscle isoforms, respectively. Since the fluorescence intensity of the probe attached to the N-terminal segment increased by about 2%–9% (*Śliwińska et al., 2011*), it appeared that the fluorophore bound to the C-terminal segment of TM was more sensitive to myosin binding. The change of fluorescence observed in this work suggested that the C-terminal

**Peer**J

**Table 5  S1-induced changes in FRET between AEDANS-labeled TMs and DABMI-labeled actin.** Conditions are as given in Table 3. Average values ±S.E. were taken from 6 to 9 independent experiments. S1/actin molar ratio is the ratio of strongly bound myosin heads required for half-maximal change in FRET. In parentheses, S1/TM molar ratio obtained by multiplying S1/actin by the number of actin subunits bound by one molecule of each TM isoform.

| TM isoform | $Q_D$ | $R_0$ | $E$ | $R$ (Å) | $\Delta R_{S1}$ (Å) | S1/actin molar ratio |
|---|---|---|---|---|---|---|
| skTM/A269C | 0.15 | 35.0 | $0.28 \pm 0.01$ | $41.0 \pm 0.1$ | 3.7 | $0.15 \pm 0.02$ (1.05) |
| TM2/S269C | 0.09 | 32.5 | $0.26 \pm 0.02$ | $38.8 \pm 0.2$ | 2.4 | $0.12 \pm 0.01$ (0.84) |
| TM5a/S232C | 0.22 | 37.6 | $0.44 \pm 0.02$ | $38.9 \pm 0.4$ | $-3.8$ | $0.08 \pm 0.02$ (0.48) |
| TM1b9a/A232C | 0.14 | 35.0 | $0.24 \pm 0.03$ | $42.6 \pm 1.2$ | $-5.5$ | $0.14 \pm 0.03$ (0.84) |

region of TM either directly interacted with myosin or significantly changed conformation upon myosin binding to actin. To explore the later possibility, changes in FRET distances in the presence of myosin were analyzed.

Strongly bound S1 shifted the C-terminal regions of all TM isoforms, but the direction of the shift observed for HMW and LMW isoforms was different. As compared to the TM-actin complex, in the presence of S1 the energy transfer efficiency between AEDANS attached to skTM or TM2 and DABMI-actin decreased. In contrast, when the energy donor was attached to TM5a or TM1b9a, an increase in transfer efficiency was observed (Table 5). Binding of unlabeled actin-S1 did not shift the maximum of the fluorescence spectrum, thus the spectral overlap ($J$) was unchanged. However, the quantum yield of AEDANS-TM bound to unlabeled actin-S1 increased (Table 5), which called for recalculation of the critical distances (see Eq. (5) in *Śliwińska et al. (2011)*). Based on the new values of $R_0$ and transfer efficiencies ($E$) obtained for AEDANS-TMs saturated with DABMI-actin-S1, donor-acceptor distances were calculated (Table 5).

The degree of the maximal shift induced by S1 binding to actin ($\Delta R_{S1}$) was calculated as the difference between the donor-acceptor distance in the absence and in the presence of S1. The data indicate that the C-terminal region of the HMW isoforms was shifted away from the donor, whereas in LMW it was shifted closer to the donor.

The S1-induced activation of the filament is a very cooperative process which means that the final change in TM orientation is achieved at S1 concentrations far below the concentrations required for actin saturation (*Eaton, 1976*; *Moraczewska, 2002*). Figure 3 shows the effects of increasing S1 concentrations on energy transfer between donor-labeled TM and acceptor-labeled actin. The experimental data was normalized and fit to the Hill equation. The ratios of strongly bound myosin heads to actin required for the half-maximal change in the energy transfer (S1/actin molar ratio) are shown in Table 5. The numbers in parentheses show the S1/TM molar ratio, obtained by multiplying S1/actin by the number of actin subunits bound by one TM molecule. The data shows that the cooperativity was very high for each of the isoforms. The differences depended on the type of end-to-end overlap.

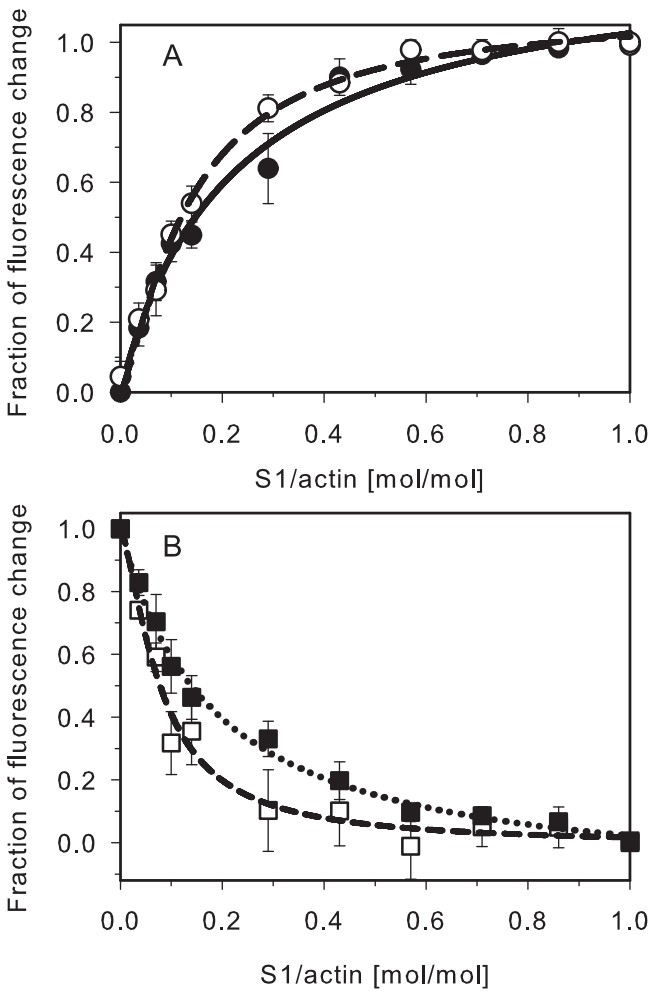

**Figure 3  Myosin S1-induced changes in fluorescence of AEDANS-TM bound to DABMI-actin.** (A) HMW TM isoforms: skTM (closed circles, solid line), TM2 (open circles, long dash line) (B) LMW TM isoforms: TM5a (open squares, short dash line), TM1b9a (closed squares, dotted line). AEDANS-TM isoforms at 0.6 μM were bound to 4.8 μM DABMI-actin and titrated with myosin S1. The data obtained for HMW isoforms was normalized by using the equation $(I − I_A)/(I_{S1} − I_A)$. For LMW isoforms the equation was: $(I − I_{S1})/(I_A − I_{S1})$, where $I$ was the fluorescence intensity at given titration point; $I_A$ was the intensity of TM-F-actin complex in the absence of S1; $I_{S1}$ was the fluorescence intensity of AEDANS-TM at maximal S1 concentration. The points were averaged from 3 to 4 independent experiments. The lines were obtained by fitting the experimental data to the Hill equation. Conditions are as described in Table 3.

## DISCUSSION

The position tropomyosin assumes on the filament controls interactions of the filament with many actin-binding proteins, thus it is an important determinant of actin filament functions. The present work is a continuation of our previous studies on the structural diversity among tropomyosin isoforms determined by sequences of the end-to-end junctions (*Śliwińska et al., 2011*). In this study, positions of the C-terminal segments of four tropomyosin isoforms relative to the actin C-terminal region were analyzed with the

use of steady-state FRET. Distances were measured between a donor specifically attached to the C-terminal segment of TM and an acceptor bound to actin's penultimate Cys374, which is located in the outer domain of the filament.

According to the NMR as well as the crystal structures of muscle and non-muscle TM model peptides, the overlap region is flexible. When forming a complex with the N-terminus, the C-terminal helices open up and interlock with the N-terminal coiled-coil (*Greenfield et al., 2006*; *Greenfield, Kotlyanskaya & Hitchcock-DeGregori, 2009*; *Frye, Klenchin & Raymont, 2010*). Although the high resolution structure of tropomyosin overlap bound to actin is not known, most probably this mode of tropomyosin end-to-end interaction along the actin filament is maintained, as it fits well into atomic models of F-actin-TM (*Barua, Pamula & Hitchcock-DeGregori, 2011*; *Li et al., 2011*). The results obtained in this work suggest that, when bound to actin, the overlap complex remains flexible. The donor-acceptor distance obtained for the isoforms with the same C-terminal sequence encoded by exon 9a (skTM and TM1b9a) or 9d (TM2 and TM5a) differed by about 6–10 Å. This shows that interdigitation of the C-terminal sequences with different types of N-termini changes conformation of the C-terminal helices by various degrees. The position of each C-terminal segment of the studied TM isoforms was not determined by the exon 9-encoded sequence, but rather by the sequence of the end-to-end overlap complex.

Interestingly, in the case of TM1b9a the donor was separated from the acceptor by a much larger distance than in the three other isoforms indicating that the non-muscle 1b-encoded N-terminus strongly distorts the structure of the striated muscle-specific 9a-encoded C-terminus.

In the earlier study, three out of the four TMs used in this work were studied: TM2, TM5a, and TM1b9a. The TMs were labeled with AEDANS in the N-terminal segment (residues 23 or 28) to measure the distance separating the donor located near the N-terminal half of the end-to-end junction and the acceptor bound to actin's Cys374 (*Śliwińska et al., 2011*). In TM2, the FRET distances obtained for donors attached to the N-terminal and the C-terminal regions were 40.2 and 36.4 Å, respectively. In the case of TM5a, the respective distances were 39.3 and 42.7 Å. Taking into account the length of the donor and acceptor probes (about 10 Å) and their random distribution, the differences of about 3.5 Å between both distances in the two isoforms were small. Thus, when bound to actin, TM2 and TM5a seem to be slightly bent within the end-to-end junction. In contrast, in TM1b9a the FRET distances measured from the N- and C-terminus were 34.8 and 48.1 Å respectively. The 13.3 Å difference suggests that the overlap of this isoform is bent or even broken. Together the results suggest that curvature of the overlap region is an inherent attribute of TM isoforms. As suggested before, bending is important to adopt the helical structure of the actin filament (*Holmes & Lehman, 2008*).

During the activation of the filament by strongly bound myosin heads, the N-terminal segments of TM isoforms were differently shifted from the positions they occupied on the filament in the absence of myosin (*Śliwińska et al., 2011*). In the present work we observed that the donor-acceptor distances measured for the C-terminal segments of

HMW isoforms (skTM and TM2) increased upon binding of myosin heads to actin, whereas in LMW isoforms (TM5a and TM1b9a) the distances decreased. This shows that the extent of the S1-induced shift of the C-terminus is determined by the type of the N-terminal sequence. However, the direction of the shift is not certain. If FRET measured a single donor-acceptor distance between AEDANS bound to TM and DABMI attached to Cys374 of only one actin subunit, shortening the distance would mean shifting the C-terminal segment towards actin's subdomain 1, where myosin-binding sites are located. This, however, would be a discrepancy compared with the earlier observations that LMW isoforms are better activators of actomyosin ATPase than HMW isoforms (*Skórzewski et al., 2009*). However, the single TM-bound donor in the TM-actin complex is surrounded by multiple acceptors attached to actin subunits, which contribute to the energy transfer with various efficiencies (*Bacchiocchi, Graceffa & Lehrer, 2004*). According to 3D reconstructions of the filaments' electron micrographs, binding of myosin to actin shifts TM azimuthally towards the inner domain of actin filament (*Lehman, Craig & Vibert, 1994*; *Lehman et al., 2000*; *Xu et al., 1999*). Such a shift might increase FRET transfer efficiency between donor and acceptors attached to actin subunits, which belong to the second chain of the long-pitch actin helix. To verify this possibility, we used atomic models of actin-TM (*Li et al., 2011*) and actin-TM-S1 (*Behrmann et al., 2012*) and analyzed changes in distances between Cys190 in the central region of TM and the five closest Cys374 residues. The cysteines were located in three actin subunits, which directly bound TM along the filament ($A_{-1}$, $A_0$ and $A_{+1}$), and in two subunits across the filament ($A_{-2}$ and $A_{+2}$). The analysis showed that in the absence of S1 Cys374 in $A_0$, $A_{-1}$ and $A_{-2}$ were the closest to Cys190. In the presence of S1 Cys190 was shifted away from Cys374 in $A_0$ and $A_{-1}$, whereas it was moved towards Cys374 in $A_{+2}$ and $A_{-2}$. Thus, in the S1-induced open state the acceptors bound to actin subunits across the filament significantly contributed to FRET efficiency. Even though the ends are missing in the actin-TM-S1 model, and in the actin-TM model the overlap is not resolved, we assume that these considerations also hold true for the end-to-end junction. Binding of S1 diminished the FRET distances of LMW isoforms, and therefore it appears that, in these isoforms, the end-to-end overlap was more strongly shifted towards actin subunits across the filament than in HMW TM isoforms.

According to the actin-TM-S1 model (*Behrmann et al., 2012*), binding of myosin heads to actin shifts TM by as much as 23 Å. The S1-induced shift which we observed in this and the previous work was much shorter, which could be explained by the multi-acceptor system discussed above as well as by local differences in TM isoforms bending.

Our data has also shown very high cooperativity of the S1-induced activation of the filament. Depending on the isoform, the maximal shift was completed when about 1–2 myosin heads per one TM molecule were bound. It is worth noting that the C-terminal segment of TM1b9a showed similar cooperativity as the C-termini of the other isoforms. However, it was much greater than the cooperativity of the N-terminal segment of TM1b9a which required about 0.4 S1/actin for half maximal saturation of the changes in FRET distance (*Śliwińska et al., 2011*). This result supports our conclusion that both ends of this chimeric TM are not compatible with each other.

## CONCLUSIONS

The FRET data gives us an insight into the dynamic changes in the positions of various end-to-end junctions. Differences in FRET distances measured between C-termini of tropomyosin isoforms and actin in closed and open activation states of the filament are determined by isoform-specific sequences. Based on the results of this and the earlier work, we conclude that the end-to-end junction of tropomyosin isoforms assume different positions on actin. The degree of TM shift in response to the filament activation by myosin is individually determined by the sequences of both ends. The results agree with the observation based on crystal structure of the TM end-to-end overlap that the intermolecular junction is flexible (*Greenfield, Kotlyanskaya & Hitchcock-DeGregori, 2009*; *Frye, Klenchin & Rayment, 2010*). Because TM isoforms are functionally specific, the present data give a structural explanation of this specificity and helps us to understand the steric and cooperative mechanisms of the thin filament regulation.

## ACKNOWLEDGEMENTS

We thank William Lehman and Edward Li for sharing the coordinates of the actin-TM model with us.

### Funding

This work was supported by a grant (N N301 723841) from the National Science Center, Poland. The funder had no role in study design, data collection and analysis, decision to publish, or preparation of the manuscript.

### Grant Disclosures

The following grant information was disclosed by the authors:
National Science Center, Poland: N N301 723841.

### Competing Interests

The authors declare that they have no competing interests.

### Author Contributions

- Małgorzata Śliwińska performed the experiments, analyzed the data, contributed reagents/materials/analysis tools.
- Joanna Moraczewska conceived and designed the experiments, analyzed the data, contributed reagents/materials/analysis tools, wrote the paper.

### Ethics

The following information was supplied relating to ethical approvals (i.e., approving body and any reference numbers):

Ministry of the Environment, Poland

Minister of the Environment for the laboratory use of genetically modified organisms: permit no. GMO:01-112/2012.

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
