# Peer review of "Structural differences between C-terminal regions of tropomyosin isoforms"

_PeerJ, doi:10.7717/peerj.181_

## Round 0.1 · original submission · Minor Revisions

· Academic Editor

Minor Revisions

You don't need to do any additional experiments but you should revise your discussion and address other reviewer's comments.

Reviewer 1 ·

Basic reporting

The article provides sufficient introduction and background to support the experimental design of the study. Formatting of Figures appears appropriate and the article seems to conform with template information.

Minor comments:
In the abstract, it should be clarified that the donor (AEDANS) is bound to the tropomyosin isoforms.
In the first paragraph of the introduction (line 18), S1 should be introduced as being the S1 subfragment of myosin which may not be clear to an audience not familiar with the field of research.
Typo in line 75: Change ‘Than cDNA from the first…, to ‘Then, cDNA from the first…’
Typo in line 186: Change ‘the donor-acceptor distance the absence’ to ‘the donor-acceptor distance in the absence’
Reference ‘Barua et al., 2013 (lines 281-283) needs volume and page information

Experimental design

The research question is clearly defined and directly links this study with the findings of the authors from a previous study. The approach and experimental design has been validated by the previous study. The research question is a logical consequence from their previous findings. The method are described with sufficient detail.

Validity of the findings

In the present study, Slivinska and Moraczewska determined isoform-specific localisation of tropomyosin C-termini along actin filaments and muscle myosin S1 subfragment-dependent changes in the localisation of the C-termini. This work is a direct follow-up study where the authors have tested the properties of the N-termini of the same tropomyosin isoforms. The current study reveals interesting information on the displacement of the C-termini of low- and high molecular weight tropomyosins in the presence of myosin S1 binding. These findings provide significant experimental evidence for an isoform specific mechanism by which different tropomyosins regulate the access of actin-associated proteins to the filaments.
The recognition of tropomyosins as ‘gate keepers’ of actin filaments is an emerging concept in the field of studying the regulation of cellular architecture. A large diversity in the regulation of actin-associated proteins by tropomyosins has particularly been explored previously for non-muscle tropomyosins. In this context it would be interesting to test differences in the S1 subfragment-dependent C-terminus displacement of tropomyosins for which isoform-specific regulation of the interaction between actin-associated proteins and the filament has been shown in different cell types. This includes for example the different properties of the TPM1 gene product TmBr3 (containing exons 1b and 9c) and the TPM3 gene product Tm5NM1 (containing exon 1b and 9d) (Bryce et al., 2003). Similarly, it would be interesting to test the effect of non-muscle myosin S1 (e.g. non-muscle myosin IIb S1) on displacement of the tropomyosin C-Termini due to the different binding properties of the muscle and non-muscle S1.

Additional comments

no comments

·

Basic reporting

The manuscript 647 by Śliwińska and Moraczewska describes the influence of the C-terminal sequence of tropomyosin (C-terminal exons 9a or 9d) on the distances between this part of Tm bound to F-actin and Cys374 of actin. For this purpose, four Tm isoforms with different C-terminal exons, 9a or 9d, were used, and the distances between fluorescence donor attached to Cys residue introduced into this Tm exon (Cys 269 or Cys 232) and acceptor attached to Cys374 of actin were measured by FRET in two different states of Tm on the surface of actin filament (closed and open).

This paper continues the previous paper of the authors (Sliwinska et al. (2011) Cytoskeleton 68, 300-312 ). Both papers are very similar (and sometimes even identical) in respect to Tm isoforms and the methods used, with the only difference that in the previous paper the fluorescence donor was attached to Cys residue in the N-terminal part of Tm, whereas in the present work it was introduced into the Tm C-terminal part. Taken together, the results of both studies show that the distances between Tm end-to-end overlap region and actin are different for Tm isoforms, and their changes in response to myosin S1 binding are individual for each isoform. Generally, I agree with authors’ conclusion that the Tm isoforms are differently oriented on actin filament and the degree of Tm shift in response to the filament activation by myosin S1 binding (i.e. transition from closed to open state) is individual for each type of the studied isoform.

Experimental design

The FRET experiments are carefully performed and clearly described, and the results are well explained.

Validity of the findings

The work is mostly novel, although the main aspects of the research have been previously published by the authors (see above).

Additional comments

n my opinion, the paper could be quite suitable for publication in PeerJ after some corrections according to the following comments.

1). First line in the abstract: “Tropomyosins are actin regulatory proteins...”
It seems to me, “Tropomyosins are actin-binding regulatory proteins..” is more correct.

2). Results, line 102: “...illustration of the overlap regions are shown in Fig. 1.”
It should be either ”...illustration of the overlap regions is shown in Fig. 1” or “...illustrations of the overlap regions are shown in Fig. 1.”

3). Table 1, upper line (exon 9a, red): it should be A/C, but not A/.

4). The English needs to be edited carefully, and the formatting of the text should be improved (e.g., on lines 87-89, 185, 193-197, and 258-263 the size of the letters is too small).

Reviewer 3 ·

Basic reporting

In this paper the authors have used FRET to study the relationship of the C terminus of tropomyosin to Cys 374 on actin in Tm isoforms that would have different overlap complexes, and the cooperativity of myosin binding. The rationale is sound: there are structures of the overlap regions of tropomyosins, and models for actin-Tm based on Tm models and structures that do not contain the overlap region. Since the ends are known to be so important for Tm function and a source of isoform diversity (work from Dr. Morazcewsaka’s lab and others), this is a reasonable approach and significant goal. Cysteine mutations were introduced just outside of the overlap region in four different Tms and shown to have little effect on the function of the Tm. The relative affinities are consistent with her previous work (Moraczewska et al., 1999) but not cited, and the fluorescence quenching is consistent with the affinities. The appropriate controls were carried out.
The FRET distances were determined for four Tms with sequences encoded by four different combinations of N-terminal and C-terminal sequences encoded by alternatively-expressed exons. Three occur naturally, the fourth is a synthetic construct (1b9a). While Tm2, Tm5a and 1b9a are all alpha Tm (TPM1) sequences, that used for the skTm is not clear: unacetylated alpha TM binds poorly; the citation is in a paper that studied gamma skTm (TPM3) that is 285 residues long due to an extra Met at the N terminus. This needs to be clarified, and also any differences from the common exons in the alpha-Tm encoded forms. The calculated FRET distances for the long forms from the donor on Cys374 on actin were smaller than for the short Tm isoforms. The cooperativity of the response from S1 binding was cooperative in all forms, but the probe in long forms moved further from the donor, while that on the short forms moved closer.

Experimental design

see above

Validity of the findings

With structures and models now available, a value of FRET studies is the opportunity to interpret the observed FRET distances in terms of these models. This is where the manuscripts falls short, and it is quite confusing at best. The data used to make actin-Tm based on 3D reconstructions do not contain information about the axial position of Tm on actin. Previously published work from the Lehrer and Miki laboratories include computational studies to try to use the FRET data in positioning Tm on actin, considering the Tm probe could have FRET with the donor on more than one actin in the filament. The models in the present paper integrate the present FRET results into a model, rather than using the data as a way to test the model. Undoubtedly the FRET data are consistent with more than one model, given the size of the FRET distances.
Cartoons in Figure 4, that derive from published models (but it is not clear which one or how), are especially confusing, it that the authors have inferred that the complex between the N and C termini of 1b9d dissociates on actin, leading to different positions of the ends on actin (i.e. no continuous cable along the actin filament), and failure to interact with the expected sites on actin, D25, K326 and K238. This is especially surprising given the published results that the 1b9a ends form a tighter complex than 1b9d (based on model peptides). The interpretation of the ends coming apart is on the basis of trying to explain the long FRET value for the C terminus, compared to the others, and the shorter value of the N-terminal receptors (previously-published work). Is there another possible interpretation? Some of the reported difficulties working with 1b9a raise concern. Also, why is there no cartoon of stTm for comparison?
The observation that the N terminus determines the direction of movement in relation to the actin donor when myosin binds is particularly interesting. With the available of a molecular model of Tm on actin in the presence of myosin (Behrmann et al., 2012), some further discussion of this result would be worthwhile.

Additional comments

Some other comments. The discussions of the structures in the PDB is a bit misleading. The Greenfield and Frye structures are very similar, despite being different genes and different methods (NMR and X-ray). The dynamics showing that the overlap complex is flexible come from the Greenfield structure and were confirmed by Frye in comparing their two structures. Also, it is the Greenfield work that showed the C-terminus opens up when forming a complex with the N terminus. Frye et al. did not determine the structure of the ends alone, and the other published structures of the C terminus from the Cohen and Maeda labs are of complexes of the C terminus with itself, in an antiparallel manner. The Murekami structure is completely unrelated to and inconsistent with those determined by Greenfield and Frye. In the overlap complexes, the C-terminal helices are not parallel; they are in the solution structure of the C terminus alone, but the chains splay apart when they form a complex with the N terminus in all the structures.
There is discussion about the myosin cooperativity, but not that myosin increases the affinity of Tm for actin (and visa versa). In 162 ff, this should be mentioned.
The paper needs to be reviewed carefully for English usage, particularly the use of articles. Also, the use of the term “orientation” is misleading. In a polar structure such as the actin filament, the first reaction is that the orientation of the ends relative to each other is variable; better to use “position”, I think that is what is suggested.
In summary, the data seem valid, but the interpretation is incomplete. Without incorporating the data into existing models in a relevant way, the value of the data significantly diminishes.

---

## Round 0.2 · accepted · Accept

· Academic Editor

Accept

Font size in some parts of the manuscript is smaller than it should be. If it is still a result of transferring the doc file to the pdf file, please, inform the production staff about this problem.